# Successive Osteosarcoma Relapses after the First Line O2006/Sarcome-09 Trial: What Can We Learn for Further Phase-II Trials?

**DOI:** 10.3390/cancers13071683

**Published:** 2021-04-02

**Authors:** Eric Thebault, Sophie Piperno-Neumann, Diep Tran, Hélène Pacquement, Perrine Marec-Berard, Cyril Lervat, Marie-Pierre Castex, Morgane Cleirec, Emmanuelle Bompas, Jean-Pierre Vannier, Dominique Plantaz, Laure Saumet, Cecile Verite, Olivier Collard, Claire Pluchart, Claire Briandet, Laure Monard, Laurence Brugieres, Marie-Cécile Le Deley, Nathalie Gaspar

**Affiliations:** 1Department of Oncology for Child and Adolescent, Gustave Roussy, Paris-Saclay University, 94800 Villejuif, France; eric.thebault@gustaveroussy.fr (E.T.); laurence.brugieres@gustaveroussy.fr (L.B.); 2Medical Oncology Department, Institut Curie, 75005 Paris, France; sophie.piperno-neumann@curie.fr; 3Biostatistics Department, Gustave Roussy Institute, 94800 Villejuif, France; tdiep.tran@gmail.com; 4SIREDO Oncology Center, Institut Curie, 75005 Paris, France; helene.pacquement@curie.fr; 5Department of Paediatric Oncology, Institut D’hématologie et D’oncologie Pédiatrique, 69008 Lyon, France; perrine.marec-berard@ihope.fr; 6Department of Tumor Pediatrics, Centre Oscar Lambret, 59000 Lille, France; c-lervat@o-lambret.fr; 7Department of Pediatric and Adolescent Unity Oncology, Toulouse University Hospital, 31300 Toulouse, France; castex.mp@chu-toulouse.fr; 8Pediatric Onco-Hematology Department, University Hospital Center of Nantes, 44093 Nantes, France; morgane.cleirec@chu-nantes.fr; 9Department of Medicine, Institut Cancerologie de l’Ouest, 44093 Nantes, France; emmanuelle.bompas@ico.unicancer.fr; 10Pediatric Hematology, Centre Hospitalo-Universitaire Charles Nicolle, 76038 Rouen, France; jean-pierre.vannier@chu-rouen.fr; 11Department of Paediatric Oncology, University Hospital, 38700 Grenoble, France; dplantaz@chu-grenoble.fr; 12Department of Paediatric Onco-Haematology, Montpellier University Hospital, 34295 Montpellier, France; l-saumet@chu-montpellier.fr; 13Department of Pediatric and Adolescent Hematogy and Oncology, Pellegrin Hospital, 33000 Bordeaux, France; cecile.verite@chu-bordeaux.fr; 14Department of Medical Oncology, Institut de Cancérologie de la Loire, Lucien Neuwirth, 42270 St Priest en Jarez, France; olivier.COLLARD@icloire.fr; 15Department of Paediatric Oncology, Centre Hospitalo-Universitaire, 51100 Reims, France; cpluchart@chu-reims.fr; 16Department of Paediatric Immuno-Hematology, Centre Hospitalo-Universitaire, 21079 Dijon, France; claire.briandet@chu-dijon.fr; 17R&D Unicancer, 75013 Paris, France; l-monard@unicancer.fr; 18Methodology and Biostatistics Unit, Centre Oscar Lambret, 59000 Lille, France; m-ledeley@o-lambret.fr

**Keywords:** osteosarcoma, recurrence/relapse, phase-II trial, metastasis, RECIST

## Abstract

**Simple Summary:**

Osteosarcoma is the most common primary malignant bone tumour in adolescents and young adults. The survival of osteosarcoma patients has not improved for four decades. The purpose was to describe first and subsequent relapses in patients from the OS2006/Sarcome-09 trial, to help future trial design. Among the 434 patients with a confirmed osteosarcoma who achieved CR1 during first line treatment, 157 patients experienced at least one relapse. The 3-year progression-free and overall survival rates were 21% and 37%, respectively. Only a quarter of the patients were included in clinical trials at first recurrence. We want to promote randomised phase-II trials in osteosarcoma relapses, with broad inclusion criteria at study entry in terms of age and disease status, and PFS as primary endpoint. Surgery/local treatment of all residual lesions should be allowed when feasible. Single-arm trial design could be used for subsequent relapses.

**Abstract:**

The purpose was to describe first and subsequent relapses in patients from the OS2006/Sarcome-09 trial, to help future trial design. We prospectively collected and analysed relapse data of all French patients included in the OS2006/Sarcome-09 trial, who had achieved a first complete remission. 157 patients experienced a first relapse. The median interval from diagnosis to relapse was 1.7 year (range 0.5–7.6). The first relapse was metastatic in 83% of patients, and disease was not measurable according to RECIST 1.1 criteria in 23%. Treatment consisted in systemic therapy (74%) and surgical resection (68%). A quarter of the patients were accrued in a phase-II clinical trial. A second complete remission was obtained for 79 patients. Most of them had undergone surgery (76/79). The 3-year progression-free and overall survival rates were 21% and 37%, respectively. In patients who achieved CR2, the 3y-PFS and OS rates were 39% and 62% respectively. Individual correlation between subsequent PFS durations was poor. For osteosarcoma relapses, we recommend randomised phase-II trials, open to patients from all age categories (children, adolescents, adults), not limited to patients with measurable disease (but stratified according to disease status), with PFS as primary endpoint, response rate and surgical CR as secondary endpoints.

## 1. Introduction

Osteosarcoma is the most common primary malignant bone tumour in adolescents and young adults [1]. Despite the first-line multimodal therapeutic strategy combining neoadjuvant and adjuvant chemotherapy, complete surgical resection of the primary tumour and metastases if feasible, and adjuvant chemotherapy [2,3,4,5,6,7], survival has not improved for four decades [8]. Patients with synchronous metastases at diagnosis, poor histological response to neoadjuvant chemotherapy and non-resectable osteosarcomas harbor high risk of relapse [4]. Relapses remain a major problem experienced by one third of patients, and present mainly as lung metastases [9]. The five-year overall survival (5y-OS) rate after relapse is dismal, below 30% [9,10,11,12]. However, some patients will survive multiple relapses and others might be cured by complete surgical resection only, in particular after a local relapse or a unique pulmonary metastatic event [13]. There is no standard treatment at relapses. The benefit associated with conventional [9,10,12] or high-dose [14] chemotherapy or radiotherapy [15] remains controversial. Despite an increase knowledge on osteosarcoma tumour cell and microenvironment characteristic [16], their complex genetic and the heterogeneous microenvironment impeded routinely used of new drug is in osteosarcoma. In the last decade, osteosarcoma phase-II trials have been disappointing with no translation into consensual phase-III trial [17]. The heterogeneity of phase-II trial designs, inclusion criteria, primary endpoint of efficacy and the absence of reliable historical population have somehow blurred the interpretation of the phase-II trials in osteosarcoma [17]. To learn from osteosarcoma relapse series to optimize phase-II trial design might help improve the success of such trials. Indeed, the benefit of new molecules has to be demonstrated with appropriate phase-II trials [18], taking into account osteosarcoma specificities including: the heterogeneous presentation, from bulky disease to complete remission (CR); the difficulties in tumour response evaluation (tumour size might not shrink even with good histological response); and the age range repartition between adult and paediatric patients. The previous retrospective series of recurrent osteosarcomas [9,10,12] did not orientate their analysis to help building further appropriate phase-II trials in this setting.

Through prospectively collected data, we describe the characteristics, treatment and outcome of first and subsequent relapses of osteosarcoma patients from the OS2006/Sarcoma09 trial [2], to further discuss phase-II trial design in that context.

## 2. Materials and Methods

### 2.1. Population

The current study is an ancillary study of the French OS2006/Sarcome-09 study [2,19,20,21], conducted in both paediatric oncology and adult sarcoma centers, between May 2007 and March 2014. This OS2006/Sarcome-09 study included a phase-III randomised trial evaluating whether zoledronate in combination with chemotherapy improves event-free survival (EFS) [2]. All newly diagnosed localised/metastatic, biopsy-proven, high-grade osteosarcoma patients, were eligible for the study regardless their participation in the randomized trial. Thus, this study could be considered as a population-based study. Appropriate ethics approval and patient/parent inform consent were obtained.

As planned in OS2006/Sarcome-09 protocol (Appendix A), first-line chemotherapy was stratified by age group and post-operative chemotherapy was stratified by main risk factors [2]. Randomised patients assigned to the zoledronate group received 10 monthly intravenous infusions [2]. The follow-up schedule was similar for all patients, for an extended period of more than 3 years.

For the current study, we analysed the first and all subsequent relapses of all patients who achieved a first complete remission, defined as the absence of macroscopic tumour by the end of treatment. Patients with progressive disease during pre-operative chemotherapy and patients with macroscopic residual tumour at any sites at the end of treatment were excluded. We also excluded from the analysis patients for whom the diagnosis of osteosarcoma was not confirmed by central review, and patients who withdrawn their consent from the study.

We report here the analyses from the first and subsequent relapses of all patients prospectively enrolled in OS2006/Sarcome-09 database who had achieved a first CR during the front-line treatment.

### 2.2. Treatment Recommendations at Relapses

Osteosarcoma relapse treatment guidelines were available during part of the study period, from the paediatric bone sarcoma group of the Société Française des Cancers de l’Enfant (SFCE, 2011) and the European Society for Medical Oncology (ESMO, 2014) [22]. Isolated lung metastases could be treated by surgery alone. Systemic treatment had to be discussed in all other situations. Local treatment (such as surgery or thermoablation) of all disease sites was recommended to reach a second CR (CR2). Amputation was proposed for isolated local relapse of the extremities. No recommendation regarding radiotherapy was given. Different Phase-I and II trials were open for osteosarcoma patients during the study period (Appendix A).

### 2.3. Data Management

Data were prospectively collected within the OS2006/Sarcome-09 database at diagnosis (patient/tumour characteristics, first-line treatments), and for all first and subsequent relapses (tumour characteristics, outcome). Relapse treatment data were retrospectively collected.

### 2.4. Statistics

Post-relapse progression-free survival (PFS) was calculated from the date of relapse until next documented disease progression or death. Post-relapse OS was estimated considering death from any cause. Survival curves were estimated using the Kaplan-Meier method. Cox models were used to evaluate prognostic factors of post-relapse PFS. The association between systemic treatment and risk of further relapse was evaluated in the subgroup of patients with a unique lung nodule. A secondary analysis was performed using the Landmark method, to consider only patients who were still free of event 2 months after the first relapse to limit the possible guarantee time bias in this observational setting.

The correlation between subsequent PFS times (PFS2/PFS1 and PFS3/PFS2, where PFSx is the time interval between (x-1)th relapse and xth failure), was investigated using a non-parametric Kendall’s Tau coefficient. Kendall’s Tau coefficient was estimated considering patients who started a treatment for the relapse and experienced a subsequent event. The growth-modulation index (GMI) was computed as the ratio PFS2/PFS1 (or PFS3/PFS2) considering all patients who started a treatment, including those alive free of progression at last follow-up. The proportion of patients with GMI > 1.3 was estimated using Kaplan-Meier method. Statistical analysis was performed using the SAS software, version 9.4 (SAS Institute, Cary, NC, USA).

## 3. Results

From 522 OS2006/Sarcome-09 patients, ten patients did not have an osteosarcoma confirmed diagnosis, three withdraw their consent and 75 patients never achieved CR1 (Figure 1). Among the 434 patients with a confirmed osteosarcoma who achieved CR1 during first line treatment, 157 patients experienced at least one relapse at the date of analysis (18 January 2018).

### 3.1. First Osteosarcoma Relapses

Patient and tumour characteristics at diagnosis of the 157 patients with first relapse are detailed in Appendix A. They were comparable to the overall population at diagnosis, except for histological response to neoadjuvant first-line chemotherapy, with over-representation of poor response (≥10% viable tumour cells) for patients who relapsed (*n* = 68/157; 43%).

The median age at first relapse was 17.5 years (range, 7.7–51.6); 90% were 12 years and older (Table 1). The median interval from diagnosis to first relapse was 1.7 years (range, 0.5–7.6). Most relapses (*n* = 135, 86%) occurred more than 1 year from diagnosis, including five metastatic relapses after 5 years (3%). Twenty patients relapsed during the 3-month period following the end of front-line treatment (13%). First relapses were mainly metastatic (*n* = 130; 83%); rarely limited to the primary tumour site (*n* = 14; 9%) or combined (*n* = 13; 8%). Metastatic sites were mainly lung in 127 patients (81%; including 88 patients with lung as unique metastatic localisation), then bone (*n* = 26, 17%), pleura (*n* = 20, 13%) and other sites (*n* = 21, 13%). Overall, 121 patients (77%) had measurable disease according to RECIST criteria (≥1 extra-osseous lesion of ≥ 1 cm). Among the 88 patients with isolated lung metastases first relapse, 39 (44%) had a unique and 49 (56%) multiple nodules. The median size of the largest pulmonary nodule was 15 mm (range, 3–130 mm). The largest nodule was ≥10 mm in 62 patients (70%).

First relapse treatments (Table 2) were: local treatment alone, either surgical resection alone (*n* = 37, 24%: one with a local relapse, 31 with lung metastases only, including 21 with a unique lung lesion, and five with other metastases), or other local treatment alone (*n* = 2, radiation therapy); systemic treatment alone (*n* = 35, 22%); systemic treatment combined with surgery (*n* = 65, 41%) or other local treatment (*n* = 16, 10%). Two patients did not receive any anti-cancer treatment.

Systemic treatment was administered to 116 patients (74%). Among the 112 informative patients, 14 received a monotherapy (12%) and 98 a multi-agent therapy (88%); 109 patients received chemotherapy (97%), combined with targeted agents in 14, and three patients received targeted agents only. Overall, 106 patients received conventional chemotherapy (95%) with different classes of molecules: alkylating agents (ifosfamide, cyclophosphamide, *n* = 65); anthracyclines (adriamycin/epirubicin *n* = 54), platinum derivatives (cisplatin/carboplatin, *n* = 57); topoisomerase inhibitors (etoposide *n* = 50); antimetabolites (methotrexate, *n* = 19); combination of gemcitabine and docetaxel (*n* = 12). Fifteen patients received high-dose thiotepa [14], 6 received a metronomic treatment only (5.4%). Targeted agents used were rapamycine (*n* = 10), zoledronate (*n* = 6), sorafenib (*n* = 2), regorafenib (*n* = 1), celecoxib (*n* = 1) and anti-IGF1R (*n* = 1). Twenty-nine patients were enrolled in phase-II trials (26%), including 25 in OSII-TTP trial [14] (Appendix A). Adolescents, aged 12–18 years at relapse, were more frequently included in a phase-II trials than younger or older patients (38% versus 15%, *p* = 0.006). The proportion of patients with either an early relapse, lung metastases or measurable disease according to RECIST criteria, were not significantly different between patients accrued or not in phase-II trials.

Overall, 107/157 patients (68%) underwent surgery, alone (*n* = 37) or combined with systemic treatment (*n* = 65), radiation therapy (*n* = 1), or both (*n* = 4). Surgical resection was complete in 75/107 operated patients (70%), incomplete in 16 and unknown in 16. Seventeen patients had radiotherapy on metastatic sites (*n* = 12) and local relapse (*n* = 2). Radiation field was unknown in three. Table 2 lists the first relapse treatments.

Post-first relapse outcome was described with a median follow-up of 4.5 years (95%CI 3.9–5.6, range 0 to 8.0 years) from the time of first relapse.

Among the 155 patients treated for the first relapse, 79 (51%) achieved a second complete remission (CR2), 51 had progressive disease (PD), 15 had residual disease and response was not evaluable in 10. Most patients who achieved CR2 had undergone surgery (76/79, alone *n* = 30 or combined with chemotherapy *n* = 46). One underwent another local treatment (thermoablation). The last two received conventional chemotherapy only.

Overall, PD or a second relapse was observed in 119 out of the 157 patients, leading to death for 100. Two patients died from secondary malignancy after the first relapse. The PFS rates at 1 and 3 years post-first relapse were 42% and 21%, respectively; OS rates at 1 and 3 years of 76% and 37%, respectively (Figure 2A). Median PFS and OS were respectively of 9.4 and 24.6 months.

Among the 76 patients who did not achieve CR2, 69 experienced PD leading to death for 62; seven patients had no subsequent event reported but a short follow-up (less than 6 months for four of them). A large proportion of patients who achieved a CR2 after first relapse treatment experienced a second relapse (48/79 patients) and died (36/48 patients); however, prolonged survival in CR2 was also observed leading to a median PFS of 17.7 months, median OS of 4.8 years, 3y-PFS of 39% and 3y-OS of 62% in this subgroup.

We identified a few factors associated with PFS after first relapse (Table 3). Patients with an early relapse had a worse PFS than those with late relapse (*p* = 0.0001 in multivariate analysis). No clear cut-off could be defined as we observed a monotonic decrease in risk associated with the time interval from the diagnosis and the first relapse. In univariate analysis, PFS differences were observed according to the age at relapse (*p* = 0.02) and whether the relapse matched measurable disease according to RECIST criteria or not (*p* = 0.048), but these differences disappeared in multivariate analysis (*p* = 0.52 and *p* = 0.45, respectively). Lastly, patients allocated to the zoledronate-arm in the front-line randomised trial appeared to have a worse outcome compared to randomised patients allocated to the control group, both in univariate (*p* = 0.04) and multivariate analysis (*p* = 0.015).

### 3.2. Unique Lung Metastasis as First Relapse

Among the 39 patients with unique lung metastasis as first relapse (Appendix A), 21 had surgery alone (54%), 15 received a systemic treatment combined with surgery and three had only systemic treatment. We did not find any factor associated with the use of systemic treatment, except a trend for a lower proportion of systemic treatment use in medical oncology compared to paediatric departments (2/10 versus 16/29, *p* = 0.07). Among these 39 patients, 26 had subsequent disease progression/relapse, leading to death for 17. One patient died from a second malignancy after the first relapse. At 1 and 3 years, PFS were 50% and 33%, OS 87% and 66%, respectively (Figure 2B). Median PFS and OS were 12.6 months and 5.0 years. We did not observe any obvious outcome difference between patients who received or not a systemic treatment as PFS curves were crossing (Figure 2C). Results were similar when we excluded patients with an early event (Landmark method with a landmark time set at 2 months).

### 3.3. Subsequent Relapses

Among the 157 patients with a first relapse, 79 achieved a CR2 from whom 48 experienced a second relapse, 15 a third relapse, and three more than three relapses (Appendix A, Appendix A). The correlation between subsequent PFS durations was poor (Figure 3). Patients with a late first relapse could have a short PFS2, while patients with an early first relapse could have a very long PFS2. The estimated Kendall’s Tau coefficient was 0.163 (95%CI, 0.043–0.28). A similar pattern was observed for subsequent relapses. In this setting of a large distribution of Growth Modulation Index (GMI), the estimated proportion of patients with GMI > 1.3 was 23.2% (se = 3.6%) at the first relapse and 28.6% (se = 8.0%) at the second relapse.

## 4. Discussion

Based on the analysis of characteristics, treatment and outcomes of 157 consecutive, unselected patients who suffered a first osteosarcoma relapse, within a population of 522 patients enrolled in the French OS2006/Sarcome-09 prospective study [2], we confirmed the dismal prognostic after a first relapse (3y-PFS = 21%; 3y-OS = 37%; median PFS and OS of 9.4 months and 24.6 months, respectively), including for those with a unique lung metastatic relapse (3y-PFS = 33%; 3y-OS = 66%). We focused our study on extracting useful information to help designing further new therapeutic phase-II trials in osteosarcoma.

Despite the difference in front-line chemotherapy regimen [23,24,25], patient/tumour characteristics at diagnosis and first relapse, and the poor outcomes after first relapse, were similar to the published series [9,10,12].

Our study presents advantages over the larger published series [9,10,12,26] usually retrospective [26], often restricted to patients under 40 years old with non-metastatic osteosarcoma of the extremities at diagnosis [10,12], using various first-line chemotherapy protocols [27], over a broader and older period. We prospectively collected within the OS2006/Sarcome-09 database not only patient and tumour characteristics, treatment and outcome at diagnosis, but also at first and all subsequent relapses. Our study population is the result of a unique French front-line trial for localised and metastatic high-grade osteosarcoma of all locations conducted over a recent 7-year period (2007–2014), for both paediatric and adult populations, likely to be representative of a large set of French osteosarcoma patients [2]. No significant difference in outcome after first relapse was observed between patients who received or not doxorubicin/cisplatin in front-line treatment. Altogether, this suggests a weak influence of front-line chemotherapy on outcome after relapse as previously reported, as soon as a multi-agent chemotherapy regimen is used [4,9,12].

With such a poor prognosis in relapsing osteosarcoma patients, therapeutic innovation is warranted in the frame of phase-II trials with two objectives: to improve patient survival, and rapidly evaluate the usefulness of a new drug or combination. This attitude might be also valid in patients with a unique lung metastasis at first relapse as their prognosis is not so good. However, in our series, only a quarter of the patients were included in clinical trials at first recurrence.

Several hypotheses might explain the low participation rate in phase-II trials:(1)The lack of dedicated phase-II trials, although more than three phase-II trials were open for osteosarcoma patients during the study period;(2)the willingness of physicians to treat patients with chemotherapy agents previously shown to have activity in osteosarcoma [6] and not already used in front-line treatment;(3)the perceived lack of attractiveness of the trial objectives, either in terms of type of compound (no new targeted or immune agent) or trial design (randomization against placebo poorly accepted, especially in paediatric population);(4)the eligibility criteria preventing participation of some patients based on their age or on the recurrence presentation (restricted to measurable disease according to RECIST criteria);(5)the lack of paediatric phase-I trials evaluating new drugs;(6)the prohibition of surgical treatment in phase-II trials, despite the importance of RC.

Changes in inclusion criteria and design of osteosarcoma phase-II trials might help increase patient access to innovative therapies, as well as optimise the possibility to quickly evaluate a potential benefit of new therapies in this rare cancer.

### 4.1. Trial Entry Criteria

The age range observed at first relapse (54% under 18 years old) push to include both paediatric and adult population in any phase-II trial, as recommended in the multi-stakeholder ACCELERATE platform [28]. In the absence of paediatric data available for a new drug or combination, accrual of adolescent from 12 years old should be proposed, under cover of PK and toxicity monitoring as recommended both in US and Europe [28,29]. Inclusion age from 12-years-old will permit to cover 90% of the osteosarcoma first relapse population. Disease presentation at first relapse in terms of measurable/non-measurable disease according to RECIST criteria [30,31,32] did not significantly influence PFS. Allowing patients with evaluable disease in addition to those with measurable disease at study entry would offer a broader access to osteosarcoma phase-II trials, without jeopardizing the results as long as the primary end-point remains PFS. Patient with minimal residual disease at study entry might be also considered although PFS will be longer, either in specific trial [17,33] or in the same trial with a specific cohort or with stratification on this criterion. This way, surgical or local treatment of all sites should be allowed, thus offering the possibility of RC without jeopardizing the statistical robustness of phase-II studies.

### 4.2. Trial End-Points and Design

Response rate according to RECIST or WHO criteria has been found insufficient to evaluate drug activity in osteosarcoma, as the tumour might not shrink despite tumour cell necrosis [34]. Osteosarcoma experts consider PFS as a more appropriate primary endpoint [35]. However, the heterogeneity of osteosarcoma relapse presentation, the diversity of therapeutic choices by absence of standard-of-care, and the diversity of PFS time points used in phase-II trials [17] do not allow reliable historical PFS rate to be taken as null hypothesis. This plead in favour of randomised phase-II trials. Others suggest single arm studies as sufficient to quickly decide on drug efficacy, based on analysis of previous negative osteosarcoma phase-II trials [33], but restricted to some populations (either measurable disease or minimal residual disease) then excluding 25% of patients with evaluable but non-measurable disease. In addition to PFS as primary end-point, secondary end-points could be evaluated, such as the response rate to systemic therapies and the CR2 achievement rate after systemic plus local therapies. Some trials are even using a co-primary endpoint combining PFS and response rate [36]. Taking advantage of the possible successive events has not been explored for trial design in relapsing osteosarcoma. Some phase-II trials already tried to evaluate the correlation between GMI and OS [37,38,39,40]. Most of them found a better OS in the population with GMI > 1.33. To our knowledge, our study is the first study evaluating the correlation between subsequent times to event in individual patients, from the first relapse. We found no correlation between subsequent times to event; consequently, we think that GMI is a questionable primary endpoint in osteosarcoma phase-II trials.

## 5. Conclusions

The analyses of the first relapses in our prospective series, reinforce our position to promote randomised phase-II trials in osteosarcoma relapses [17], with broad inclusion criteria at study entry in terms of age and disease status (either evaluable or measurable disease according to RECIST criteria or minimal residual disease), and using PFS as primary endpoint. Surgery/local treatment of all residual lesions should be allowed when feasible. Single-arm trial design could be used for subsequent relapses.

## Figures and Tables

**Figure 1 cancers-13-01683-f001:**
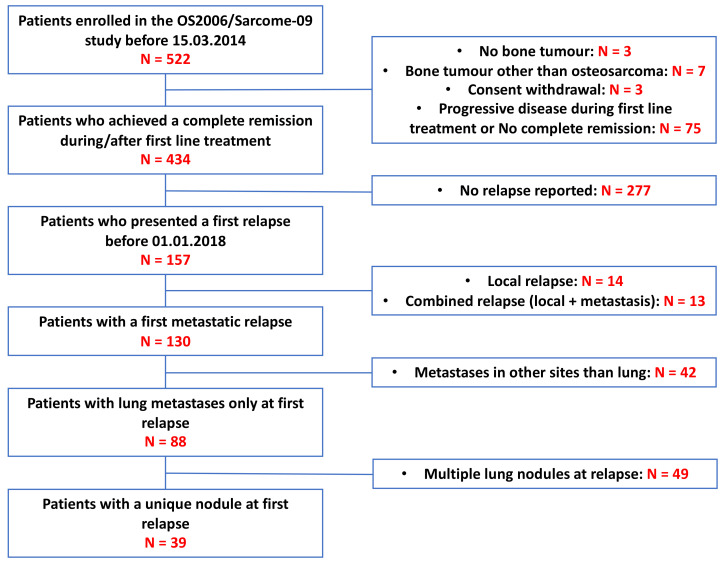
Participant flow of the current study.

**Figure 2 cancers-13-01683-f002:**
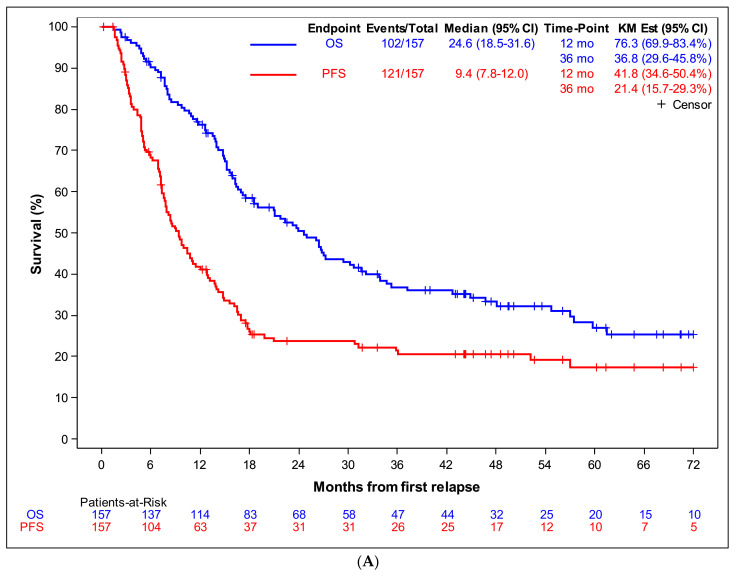
(**A**): Outcome after first osteosarcoma relapse. Overall (blue) and progression-free survival (red) of the whole cohort of 157 patients. (**B**). Outcome after first osteosarcoma relapse: Overall (blue) and progression-free survival (red) of the 39 patients with a unique lung lesion at first relapse. (**C**). Outcome after first osteosarcoma relapse: Progression-free survival of the 39 patients with a unique lung lesion at first relapse, according whether they received (blue) or not (red) a systemic treatment (chemotherapy).

**Figure 3 cancers-13-01683-f003:**
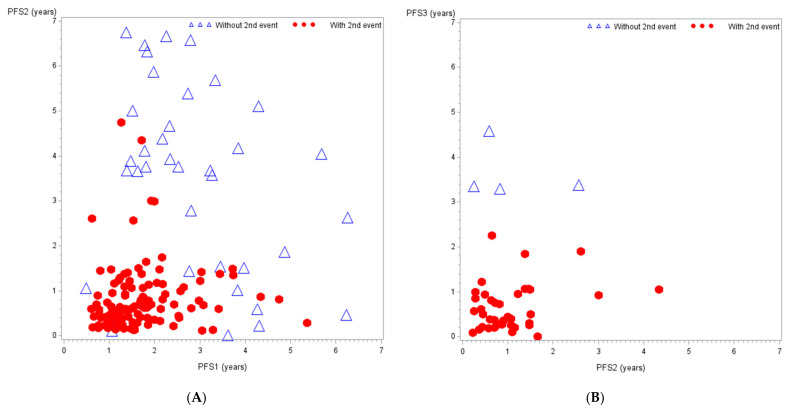
Correlation between successive PFS. (**A**). Correlation between PFS2 and PFS1. Kendall’s Tau coefficient = 0.163 (95%CI, 0.043–0.28). GMI > 1.3 = 23.2% (se = 3.6%). (**B**). Correlation between PFS3 and PFS2. Kendall’s Tau coefficient = 0.088 (95%CI, −0.17–0.33). GMI > 1.3 = 28.6% (se = 8.0%). Blue triangle: without a second event—Red dot: with a second event.

**Table 1 cancers-13-01683-t001:** Characteristics of patients and tumour at first relapse (157 patients).

Characteristics of the First Relapse	N (%) or Median (Range)
Gender	
Male, N (%)	85 (54.1%)
Female, N (%)	72 (45.9%)
Age at relapse (years)	
Median (range)	17.5 (7.7–51.6)
Less than 12 years, N (%)	16 (10.2%)
12–17 years, N (%)	69 (42.9%)
18–25 years, N (%)	52 (33.1%)
>25 years, N (%)	20 (12.7%)
Time interval from diagnosis to 1st relapse (years)	
Median (range)	1.7 (0.5–7.6)
<1 year, N (%)	22 (14.0%)
1–1.5 years, N (%)	40 (25.5%)
1.5–2 years, N (%)	40 (25.5%)
≥2 years, N (%)	55 (35.0%)
Site of relapse	
Local relapse, N (%)	14 (8.9%)
Metastases only, N (%)	130 (82.8%)
Combined (local relapse + metastases), N (%)	13 (8.3%)
Metastatic sites (possible combined)	
Lung, N (%)	127 (80.9%)
Unilateral	74
Single nodule	44
Multiple nodule	30
Bilateral	53
Bone, N (%)	26 (16.6%)
Pleura, N (%)	20 (12.7%)
Other metastases, N (%)	21 (13.4%)
Relapse with measurable disease matching RECIST criteria, N (%)	
No, N (%)	36 (22.9%)
Yes, N (%)	121 (77.7%)
Treatment allocated by randomisation	
No randomisation, N (%)	58 (36.9%)
Without Zoledronate, N (%)	45 (28.7%)
With Zoledronate, N (%)	54 (34.4%)

**Table 2 cancers-13-01683-t002:** Treatment administered for the first relapse.

Treatment for Relapse	N (%) or Median (Range)
Treatment modalities	
Systemic treatment, N (%)	116 (73.9%)
Surgery, N (%)	107 (68.2%)
Radiation therapy, N (%)	17 (10.8%)
Thermoablation, N (%)	2 (1.3%)
Other ^(1)^, N (%)	1 (0.7%)
Treatment combination	
No treatment, N (%)	2 (1.3%)
Local treatment alone:	
Surgery alone, N (%)	37 (23.6%)
Other local treatment ^(2)^ +/− surgery N (%)	2 (1.3%)
Systemic treatment alone, N (%)	35 (22.3%)
Systemic treatment + Surgery, N (%)	65 (41.4%)
Systemic treatment + other local treatment ^(2)^ +/− surgery, N (%)	16 (10.2%)
Details of systemic treatment (N = 112, 4 missing data)	
Type of molecules	
Chemotherapy, N (%)	109 (97.3%)
Targeted agent ^(3)^, N (%)	17 (15.2%)
Targeted agent + chemotherapy	14 (12.5%)
Targeted agent alone	3 (2.7%)
Combination	
Monotherapy, N (%)	37 (33.0%)
Multi-agent therapy, N (%)	75 (67.0%)
Treatment scheme	
Conventional courses only, N (%)	85 (75.9%)
Conventional courses followed by HD chemo ^(4)^, N (%)	12 (10.7%)
Metronomic treatment only, N (%)	6 (5.4%)
Conventional courses & metronomic treatment, N (%)	6 (5.4%)
Conventional courses + HD chemo ^(4)^ & metronomic treatment, N (%)	3 (2.7%)
Participation in a clinical trial	29 (25.9%)

^(1)^ Other: samarium 153. ^(2)^ Other local treatment: radiation therapy (N = 16), thermoablation (N = 1), both (N = 1). ^(3)^ Targeted agents included: rapamycin (N = 10), zoledronate (N = 6), sorafenib (N = 2), regorafenib (N = 1), celecoxib (N = 1) and anti-IGF1R (N = 1). Only three patients received targeted agents without chemotherapy: zoledronate (49.6 years), regorafenib (49.4 years) and anti-IGFR1 (17.0 years). ^(4)^ HD chemo: high dose chemotherapy (thiotepa) followed by autologous stem cell graft.

**Table 3 cancers-13-01683-t003:** Prognostic factor study of the progression free survival after first relapse.

Factors	N	Univariate Analysis			Multivariate Analysis ^(1)^		
Patient and Tumour Characteristics	N	HR	95%CI	*p*-Value	HR	95%CI	*p*-Value
Metastasis at diagnosis				0.20			‒
No	133	1					
Yes	24	1.38	(0.86–2.21)				
Chemotherapy as first-line treatment				0.65			‒
Methotrexate-etoposide-ifosfamide	124	1					
API-AI	33	1.11	(0.71–1.72)				
Histological response of primary tumour				0.26			‒
Good histological response	89	1					
Poor histological response	68	1.23	(0.86–1.76)				
Treatment allocated by randomisation ^(2)^				0.040			0.015
Without Zoledronate	45	1			1		
With Zoledronate	54	1.61	(1.02–2.53)		1.79	(1.12–2.86)	
Age at relapse (years)				0.02			0.52
Less than 12 years	16	1			1		
12–17 years	69	1.43	(0.78–2.62)		1.26	(0.47–3.38)	
18–25 years	52	0.78	(0.41–1.48)		0.84	(0.30–2.32)	
>25 years	20	1.52	(0.73–3.17)		1.26	(0.47–3.38)	
Time interval from diagnosis to 1st relapse				<0.0001			0.0004
<1 year	22	3.92	(2.23–6.91)		4.54	(2.13–9.68)	
1–1.5 year, N (%)	40	2.44	(1.50–3.95)		2.87	(1.53–5.38)	
1.5–2 years, N (%)	40	1.64	(1.00–2.68)		1.76	(0.91–3.38)	
>2 years	55	1			1		
Type of relapse							
Local relapse	14	1		0.32			‒
Unique lung metastasis	39	0.72	(0.36–1.45)				
Other	104	1.00	(0.53–1.88)				
Measurable disease matching RECIST criteria							
No	36	1		0.048	1		0.45
Yes	121	1.57	(0.99–2.46)		1.27	(0.68–2.39)	

Prognostic factor study of the progression free survival after first relapse. Univariate and multivariate ^(1)^ analysis was performed on the whole population of 157 patients who experienced at least one relapse, except for the “treatment allocation by randomisation” ^(2)^ performed on the OS2006 randomised population of 99 patients (the 58 patients non-randomised were excluded). The results of the multivariate ^(1)^ analysis displayed in the table are based on the model including the four variables: treatment allocation by randomisation; age at relapse in 4 categories; time interval from diagnosis to first relapse in 4 categories; and measurable disease at relapse, matching criteria.

## Data Availability

The data presented in this study are available on request from the corresponding author. The data are not publicly available.

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
