# Peer review of "Successive Osteosarcoma Relapses after the First Line O2006/Sarcome-09 Trial: What Can We Learn for Further Phase-II Trials?"

_cancers, 2021, doi:10.3390/cancers13071683_

Round 1

Reviewer 1 Report

The authors present a rather large cohort of osteosarcoma patients experiencing first relapse. The patients are a subgroup recruited at first relapse from a larger front-line trial. The study seems well conducted and will provide much needed knowledge on the management of recurrent osteosarcoma. However, some issues needs to be resolved before publications.

The motivation of the study needs to be further defined and the discsussion enhanced to answer these questions.

Languange: Overall language quality have to improve. The manuscript should be proofed by a person native in the English language.

Introduction: Is short and to the point. Could be expanded, but not crucial.

MM  - population: Inclusion and exclusion criteria should  be clearly stated. Patients in the randomized study should be presented separately from patients enrolled outside of the study. Why were 75 patients that did not reach CR1 excluded from the current study?

Results
Table 3: It is stated that "This model includes 99 observations because the   who did not participate in the randomised trial are excluded". Where patients come from should be more clearly stated as pointed out above.

Figure 3: Has very small labels and is therefore hard to read. Colors should have better contrast.

Reviewer 2 Report

The article is interesting in its field and requires a minor revision before the acceptance.

The Introduction section is really poor and it has to be expanded.

Author Response

The article is interesting in its field and requires a minor revision before the acceptance.
The Introduction section is really poor and it has to be expanded.

The introduction has been expanded, as requested by reviewer 1 and 2, to clarify the purpose of the study without added to much text to keep the requested number of words by the journal.
Some reference has been added and the numbers changed accordingly.

Reviewer 3 Report

There is a minor discrepancy between the accurte description of the targeted therapy, used for relapse treatment, and the description of the chemotherapy (rules 183-184). I would recommend the authors to be more clear and give more information about the chemotherapy that was applied, like agents and dose. 

  • typo rule 231: p    0.0001 ?? must be p = 0.0001?
  • typo rule 335: in term of -->in terms of.

Author Response

There is a minor discrepancy between the accurate description of the targeted therapy, used for relapse treatment, and the description of the chemotherapy (rules 183-184). I would recommend the authors to be clearer and give more information about the chemotherapy that was applied, like agents and dose.

Page 6: Systemic treatment was administered to 116 patients (74%). Among the 112 in-formative patients, 14 received a monotherapy (12%) and 98 a multi-agent therapy (88%); 109 patients received chemotherapy (97%), combined with targeted agents in 14, and three patients received targeted agents only. Overall, 106 patients received conventional chemotherapy (95%) with different classes of molecules: alkylating agents (ifosfamide, cyclophosphamide, n=65); anthracyclines (adriamycin/epirubicin n=54), platinum derivatives (cisplatin/carboplatin, n=57); topoisomerase inhibitors (etoposide n=50); anti-metabolites (methotrexate, n=19); combinaison of gemcitabine and docetaxel (n=12). Fifteen patients received high-dose thiotepa [14], 6 received a metronomic treatment only (5.4%).

typo rule 231: p 0.0001 ?? must be p = 0.0001?
typo rule 335: in term of -->in terms of.

Typo problems have been corrected.